# Sustainable Shopping Mall Rehabilitation

**Mu-Fa Lin \*** [ID]**, Shen-Guan Shih and Yeng-Horng Perng**

Department of Architecture, National Taiwan University of Science and Technology (NTUST), 43, Section 4, Keelung Road, Taipei City 106, Taiwan; sgshih@mail.ntust.edu.tw (S.-G.S.); perng@mail.ntust.edu.tw (Y.-H.P.)
* Correspondence: d10613013@gapps.ntust.edu.tw

**Abstract:** In the era of globalization, increasing the nation's industrial competitiveness, production value, and competitiveness is a challenge for every country. Therefore, Taiwan must strive for innovation, increase the added value of its products, and enhance industrial competitiveness. This promotes Taiwan's survival under competition engendered by globalization. The study identified public places that influence consumer opinions of shopping malls to examine the intentionality of the relationship between people and shopping malls. Taiwan has many historical cultural parks that have been reduced to idle spaces. This research will enable historical creative parks. Through DAHP's research methods, this research will determine the key factors in the public space of cultural shopping centers. Moreover, the study examined key factors that create a sense of culture in public space design. The relationship between the general public and shopping malls has various internationalities.

**Keywords:** sustainable; refurbishment; interior design; culture shopping mall space

---

## 1. Introduction

Today, shopping malls are not developed for a single functional purpose [1,2]. The contemporary shopping mall demonstrates diverse manifestations of creativity and culture. Relative to the shopping venues of the past, today's shopping malls have more complex and rich cultural functions [1]. The functional aspect of shopping malls serves to satisfy the economic model that connects commercial entities (tenants) and visitors (buyers) [3]. Another function is to provide a fascinating place that presents visitors with a culturally creative and unique impression through its architectural and interior design [4]. As an experiential place, a shopping mall must have a unique and special space that is suitable for socialization. To attain such functions, the spaces of shopping malls must not be placeless [2]. This study aimed to identify public spaces that influence consumers' impressions of a shopping mall to achieve two goals.

The first goal was to examine the intentionality of the relationship between people and shopping malls. The second goal was to examine key design factors that imbue public space with a sense of local culture. The relationship between the general public and shopping malls has various internationalities [5]. This study used an in-depth interview method to survey interior decoration project service providers and experts. We conducted a literature review and consulted persons with practical experience. We examined cases of work on professional interior decoration projects for cultural shopping malls. This was to create a framework for project management indicators and conduct weight analysis of each indicator for evaluation. The evaluation can serve as a reference by which service providers can allocate resources reasonably and develop strategies based on evaluation weight [6].During the study, we invited experts, such as managers of renowned cultural shopping malls in Taiwan, managers at internationally renowned brand stores, subcontractors that actually provided the construction service of interior decoration, design companies, and architects to participate in the study and to provide their valuable experience and opinions. This study adopted a Delphic hierarchy

process to extract the framework of evaluation indicators used by cultural shopping mall designers when they select their design projects [7]. We found the weight distribution of each evaluation indicator through verification of data and operational analysis [8]. Taiwan has many historical cultural parks that have been reduced to idle spaces. This research will enable historical creative parks [9]. Through DAHP's research methods [10], this research will determine the key factors in the public space of cultural shopping centers. Criteria listed in Figure 1. Therefore, the objectives of this study are to:

1. Discuss evaluation indicators of interior design projects for cultural business centers.
2. Construct a hierarchical framework of design project of interior decoration.

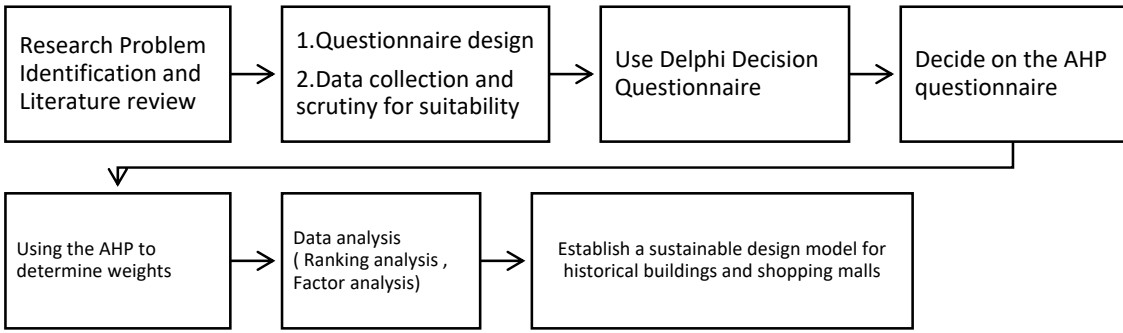

**Figure 1.** Research framework for the study.

## 2. Materials and Methods

This study invited a total of 30 experts for a questionnaire survey that used the Delphi method. We had a sample of 30 expert respondents Professors of National University, including 1 Honorary Professor of Architecture and 1 Corporate Chair Professor of School of Management [11]. There are 10 actual operators in the cultural shopping mall industry, with more than 15 years of architectural design background, and 10 people with historical and cultural reuse experience, and more than 15 years of interior design engineering background, and 8 people with actual implementation of historical and cultural mall interior design experience,30 people in total. Their background is related to construction-related industries and shopping mall operations. The questionnaire return rate of this study was 100%. In the context of the reliability and validity of the questionnaire survey, the mean of opinion was less than 15% in the first round of this study. Criteria listed in Table 1.

**Table 1.** Demographic information on valid questionnaire respondents under the modified Delphi method.

| Attribute | Category | Total Number of People | Percentage |
|---|---|---|---|
| Sex | Male | 16 | 53.00% |
| | Female | 14 | 47.00% |
| Age | 30–40 | 9 | 30.00% |
| | 41–50 | 11 | 37.00% |
| | 51–60 | 7 | 23.00% |
| | ≥60 | 3 | 10.00% |
| Educational Attainment | High school or below | 4 | 13.00% |
| | College | 8 | 27.00% |
| | University | 11 | 37.00% |
| | Graduate school or above | 7 | 23.00% |

**Table 1.** *Cont.*

| Attribute | Category | Total Number of People | Percentage |
|---|---|---|---|
| Occupation | Building interior designer | 8 | 27.00% |
| | Building material manufacturer | 2 | 7.00% |
| | Construction worker | 1 | 3.00% |
| | Professor in a Department of Interior Design | 10 | 33.00% |
| | Client with experience of interior design | 7 | 23.00% |
| | Professors of National University, including | 2 | 7.00% |

*Research Scope of Shopping Mall*

Regarding the relationship between people and local culture, each consumer must have different feelings, a unique depth of cultural atmosphere, and a meaningfulness level [1]. Suggested that the environment of shopping malls can influence shopping habits. Place attachment can be researched in shopping malls [5]. Interpersonal relationships are created by the physical and social factors that make a place [12], and these factors would subsequently influence the place's characteristics [1]. Shopping malls must create their sense of local culture through various approaches [13]. The sense of local culture is created by physical and social factors [14]. The sense of place can be attained by constructing cultural spatial forms that are appealing. The social sense of place can be triggered by organizing various social and promotion activities. On 17 December 2017, proceebdings were recorded for the second seminar of the twenty-ninth architectural research report of the Architectural Institute of Taiwan [15].

Research on architectural types of museums constructed in the United States during the museum boom before the Great Depression of the 1930ssuggests that the most crucial part of interior design is that the target being displayed must be clearly visible. Therefore, the lighting problem and the circulation problem are crucial problems [15]. The circulation of museums must flow freely and must not interfere with people who are viewing or studying artworks. Guadet also observed a critical feature of shopping malls at the time was that architecture and architectural decorations should not interfere with displays of objects. Exhibition rooms that are specifically designed for shopping malls should have special features that set them apart. The function of a palace was changed to exhibition space for a shopping mall. In this case, differences exist between the magnificent decoration of the palace and the simplicity of exhibition space designed specifically for the shopping mall [16].In each city, in order to protect cultural heritage, it is necessary to understand the relevant cultural assets of each region, as well as their use and influence in the specific environment and the entire society. Most importantly, one must consider the potential and advantages that assets provide in creating individual and collective value and experience, and their contribution to the sustainable development of cities. The form of governance must be analyzed in order to provide decision-makers and managers with information, knowledge, and tools to improve the design, implementation, and evaluation of strategies and actions, which can include cultural heritage. Therefore, social and cultural values and the collective drive of identity need to undergo a series of social changes, especially in terms of providing knowledge about the evolution of cultural identities and common values in the process of change. Finally, I aim to prove cultural heritage Sustainability [9].

This study focuses on the adaptive reuse of cultural heritage, specifically as museums or business exhibition spaces. When a historic building is transformed into a museum, problems related to its restoration and maintenance, as well as the functions, operations, and management of the museum, may occur [15].

The sense of local culture is created by physical and social factors. The sense of place can be attained by constructing cultural spatial forms that are appealing. The social sense of place can be triggered by organizing various social and promotion activities. Another strategy adopted by shopping center is multiple combinations of stores. This strategy is primarily used to control which stores can open in a shopping center in order to attract a specific type of consumer [5]. Note that the design of a shopping center must consider the following factors: 1. Innovation, 2. Uniqueness, 3. Easily maintainable, 4. Maintenance cost, and 5. Modularize ability of a space.

## 3. Results

### 3.1. Delphi Method

1. Before the birth of Christ, the word Delphi originated from the name of a famous Greek city. Legend has it that the God of sun, Apollo Pythios, used to live on Omphalos stone of the Sanctuary of Apollo in this city. The city of Delphi was famous for the oracular pronouncement of the priestesses of Apollo. Therefore, the word Delphi refers to prophecy and prediction. In modern times, Delphi is also the name of a tool used to predict the future.

2. In the early1950s, the Rand Corporation in the United States invented the Delphi Technique. The Rand Corporation conducted a national defense research effort named Project Delphi under the sponsorship of the United States Air Force. This project used a series of extensive and intensive questionnaires to understand expert opinion and used controlled feedback as an auxiliary. The project sought to understand the viewpoint of Soviet Union strategic designers. The developers attempted to identify the targets that Soviets might attack to paralyze American material flows. To minimize supply consumption, the minimal number of required atomic bombs was estimated. Finally, consistent opinions of experts were compiled to develop tactics and strategies to combat the Soviet Union.

3. In the mid-1960s, Helmer and Gorden published a paper titled "Report on a Long-Range Forecasting Study". This was the first use of the Delphi method outside of military planning. Following this, the industrial and business sectors gradually started to apply the Delphi method to cope with opinions in various types of affairs. Government agencies and academic institutions have also applied this method as a policy evaluation method. In the 1970s, the Delphi method started to be applied and tested for medical and health care purposes [17].

### 3.2. The Delphi Questionnaire Design, Data Management, and Statistical Analysis

(1) Questionnaire design: one considerable advantage of a Delphi forecasts it has no need for advanced and complex statistical methods. Therefore, the method is not only used in academia, but also used in the business sector. A typical Delphi survey questionnaire adopts four methods: 1. Open-ended questions: Open-ended questions are usually used in the first round of the questionnaire survey. A small group of researchers uses the open-ended method to encourage experts to express their opinions about a research topic freely. The purpose of this brainstorming is to elicit experts' thoughts without limiting the scope of discussion. Following the first round, researchers categorize the expert opinions through content analysis and develop a questionnaire for the second round. 2. Yes-no binary method: Following the second round, different evaluation methods were used according to the desired results. Yes-no is a type of binary inquiry that mainly asks experts to express opinions such as "agree" or "disagree," "should" or "should not," and" may" or "may not", the method is similar to voting by ballot. 3. Ranking method: This method asks experts to classify research topics into different levels based on attitudes and opinions to define each topic's importance. 4. Likertscale: The Likert scale was first developed by Rensis Likert and is also referred to as the Likert summative scale. Its design is to measure a series of attitude related opinions on a 5-point Likert scale that indicates the degree to which one holds such opinion (A Likert scale was adopted in this study).

## 4. The Analytic Hierarchy Process

(1) The analytic hierarchy process (AHP) was developed by a University of Pittsburgh professor Thomas L. Saaty in 1971 for the United States Department of Defense to conduct response plan problem research. Specifically, the AHP is used in situations with uncertainty and decision-making problems with multiple evaluation criteria. In 1972, research on rational distribution of electric power in each industry was conducted under the sponsorship of the National Science Foundation [18]. Saaty started to create scales for AHP-related judgments when he was investigating the effect of "no peace and no war" on Egyptian economic, political, and military situations for the Egyptian government in July 1972. The AHP theory gradually matured when Saaty applied it to transportation research for Sudan in 1973. The theory became more comprehensive and complete after repeated applications, modifications, and verifications from 1974 to 1978. In 1980, Saaty organized the AHP theory into a book. Subsequently, the theory has been applied in various domains such as business, engineering, and public policy [19].

(2) The AHP can be applied in uncertain situations such as multi-object and multiple evaluation criteria decision problems. Through organization of expert opinions, complex evaluation problems are analyzed into simple factor levels. Subsequently, through scale measurements, the data are quantified and used to form a pair wise comparison matrix at each level. The eigen values are obtained after calculation and each eigenvalue is used to assess the strength of each pairwise comparison matrix, which serves a reference when making decisions [20].

(3) The purpose of developing the basic hypothesis, limited AHP of the AHP was to systematize complex problems. Hierarchical decompositions were conducted from different dimensions. The context was obtained through quantified judgment and comprehensive evaluation was conducted subsequently. This provides decision-makers with adequate information for selecting appropriate plans and simultaneously reduces the risk of decision error.

### *The Introduction of AHP*

The AHP can be applied in uncertain situations such as multi-object and multiple evaluation criteria decision problems. Through organization of expert opinions, complex evaluation problems are analyzed into simple factor levels. Subsequently, through scale measurements, the data are quantified and used to form a pairwise comparison matrix at each level. The eigenvalues are obtained after calculation and each eigenvalue is used to assess the strength of each pairwise comparison matrix, which serves a reference when making decisions [8]. Criteria listed in Table 2.

**Table 2.** Table of evaluation scales of the AHP and their explanations.

| Evaluation Scales | Definition | Explanation |
| --- | --- | --- |
| 1 | Equally important | Two activities have equal contribution to the objective |
| 3 | Slightly important | One activity is slightly favored by experience and judgment over another. |
| 5 | Quite important | One activity is strongly favored by experience and judgment over another. |
| 7 | Strongly important | Extremely strong preference for one activity over another is demonstrated in practice. Strongly important |
| 9 | Absolutely important | Ample evidence is provided to confirm preference for one activity over another |
| 2, 4, 6, 8 | Intermediate values between two neighboring judgments | When compromise is necessary |

## 5. Data Analysis Methods

In this study, we used the Delphi method and AHP method to analyze key criteria of the space sustainable shopping mall rehabilitation. The indictors of the two methods' questionnaires are the same, but the approaches are different.

*Expert Questionnaire Survey through the Delphi Method*

Dalkey [11] suggested that if the experts participating in a questionnaire survey are homogenous, 15 to 30 experts are required. If the participating experts are heterogeneous, only 5 to 10 of them are required. The experts that were invited to participate in this study included a total of 30 experts for a questionnaire survey that used the Delphi method. We had a sample of 30 expert respondents Professors of National University, including 1 Honorary Professor of Architecture and 1 Corporate Chair Professor of School of Management. There are 10 actual operators in the cultural shopping mall industry, with more than 15 years of architectural design background, and 10 people with historical and cultural reuse experience, and more than 15 years of interior design engineering background, and 8 people with the actual implementation of the historical and cultural mall interior design experience, 30 people in total. Their background is related to construction-related industries and shopping mall operations. The questionnaire return rate of this study was 100%. The analysis of the results of the first expert questionnaire indicated that some experts provided explanations for some elements of indicators. That is, experts provided some suggestions to change the names of elements and dimensions slightly. In the context of the reliability and validity of the questionnaire survey, the mean of opinion was less than 15% in the first round of this study. The overall stability of the questionnaire already reached the standard and it was not necessary to conduct a second round of expert questionnaires [11].

The returned Delphi method questionnaires had Standard Deviation (SD) = 0.772 and Absolute value |MO-M| = 0.494 for each evaluation indicator. This indicated that expert opinion tended toward consistency. To meet the need for questionnaire reliability and validity, we modified an explanation column of indicators according to some experts' opinion. Compiled opinions on the first round indicated that the experts' opinions tended toward consistency. Analysis of questionnaire data indicated that expert opinion's SD was smaller than 1. Consistency testing indicated that experts' opinions |MO-M| was smaller than 1 and no |MO-M| value of greater than 1 or divergence in expert opinion was found. Criteria listed in Table 3.

**Table 3.** Delphi method questionnaire data reference list.

| NO | Dimension | Expert Code/Evaluation Criteria | Total | Mean | SD | Absolute Value |MO-M| |
|----|-----------|--------------------------------|-------|------|------|----------------------|
| 1 | | Cultural heritage connection | 138 | 4.60 | 0.6747 | 0.400 |
| 2 | Value | Adjacent unit connection | 133 | 4.43 | 0.7279 | 0.567 |
| 3 | creation | External resource import | 134 | 4.47 | 0.7761 | 0.533 |
| 4 | | Create new source of revenue | 133 | 4.43 | 0.7739 | 0.567 |
| 5 | | Appropriation control | 131 | 4.37 | 0.7649 | 0.633 |
| 6 | | Schedule management | 136 | 4.53 | 0.7303 | 0.467 |
| 7 | Economic | Easily maintainable | 134 | 4.47 | 0.7761 | 0.533 |
| 8 | advantages | Low operating cost | 133 | 4.43 | 0.7279 | 0.567 |
| 9 | | Modularity easily adjustable | 124 | 4.13 | 0.7303 | 0.133 |
| 10 | | Multiple-lease potential | 123 | 4.10 | 0.7589 | 0.100 |
| 11 | | Firm's experience | 129 | 4.30 | 0.8769 | 0.700 |
| 12 | Designers' | Elegant presentation | 122 | 4.07 | 0.9803 | 0.933 |
| 13 | concerns | Familiarity with laws and regulations | 134 | 4.47 | 0.6288 | 0.533 |
| 14 | | Excellent word-of-mouth | 130 | 4.33 | 0.8841 | 0.667 |
| 15 | Environmental | Green building materials | 127 | 4.23 | 0.7279 | 0.233 |
| 16 | consciousness | Energy conservation | 128 | 4.27 | 0.7849 | 0.733 |
| 17 | | Water conservation | 123 | 4.10 | 0.8030 | 0.100 |
| | | | | | **0.772** | **0.494** |

## 6. Steps of Conducting the AHP

Analytical Hierarchy Process (AHP) is a set of studying methods to systematize complex problems. It is suitable to assist decision-makers in choosing the appropriate program that can be hierarchically analyzed from different levels and through quantitative methods to find out the trails for making the integrated evaluation. The questionnaire return rate of this study was 90.5% [21].

The operational process of the AHP in this study is illustrated in Criteria listed in Figure 2:

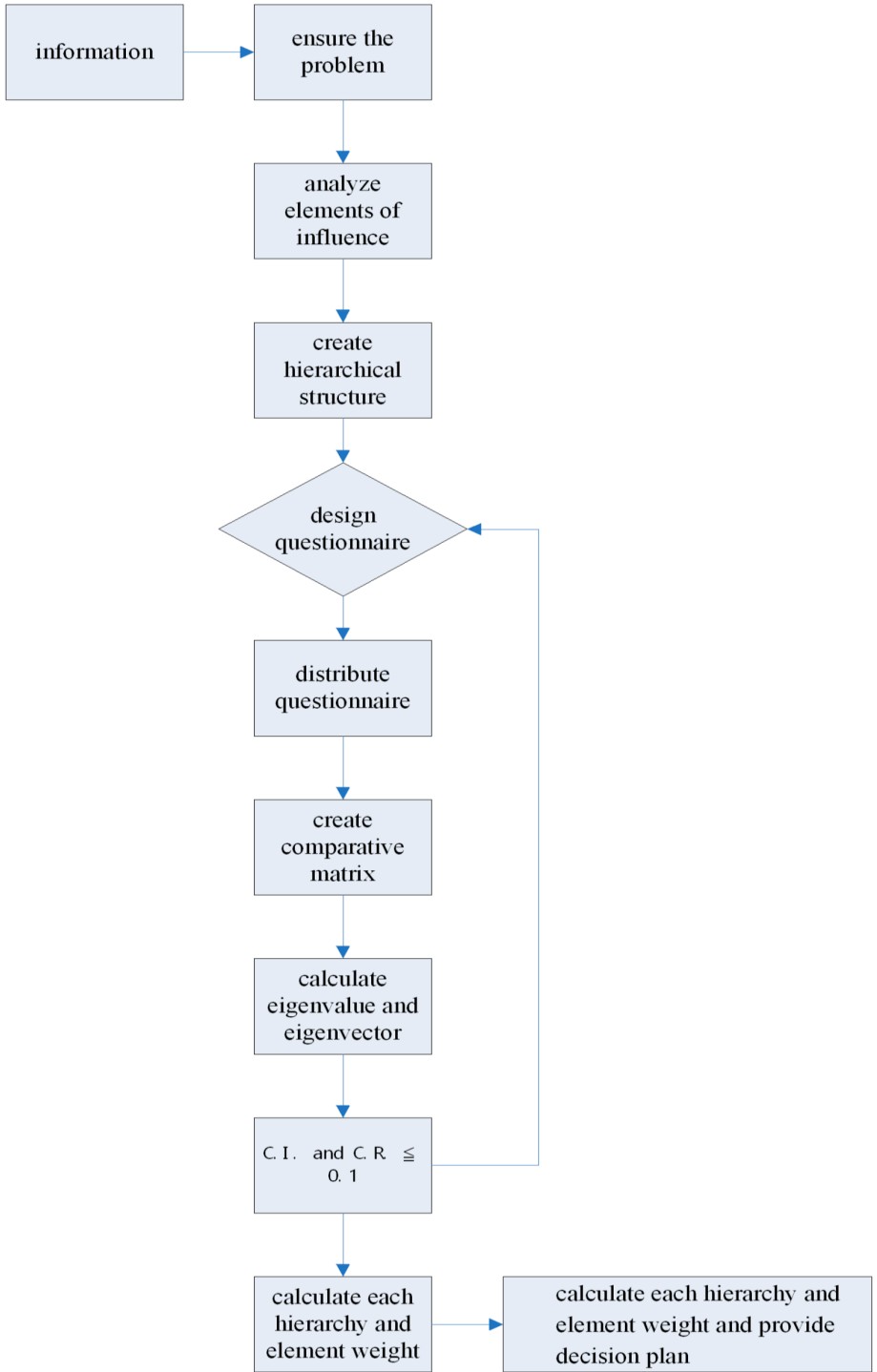

**Figure 2.** AHP operating procedures [22].

*The AHP Questionnaire*

The questionnaire return rate of this study was 90.5%. A total of 400 questionnaires were delivered, 362 were retrieved for a response rate of 90.5%. Among these 362 questionnaires, 35 were invalid and 327 were valid. The important percentage in this study is "Firm's experience" around 11.12%, second most is "Green building materials", following is "Elegant presentation". The smallest proportional factor is "Multiple-lease potential". The comparison between "Firm's experience" and "Green building materials" is very close which shows that both "Firm's experience" and "Green building materials" are important. Criteria listed in Table 4.

**Table 4.** AHP questionnaire data reference list.

| Criteria | Level (1) *Wi* | Sub-Criteria | Level (2) *Wi* | Overall *Wi* | Overall Sequence |
|---|---|---|---|---|---|
| Value creation | 31.45% | Cultural heritage connection | 30.84% | 7.71% | 5 |
| | | Adjacent unit connection | 30.90% | 7.72% | 4 |
| | | External resource import | 25.80% | 6.45% | 9 |
| | | Create new source of revenue | 12.44% | 3.11% | 13 |
| Economic advantages | 32.34% | Appropriation control | 26.38% | 6.59% | 8 |
| | | Schedule management | 25.77% | 6.44% | 10 |
| | | Easily maintainable | 11.91% | 2.97% | 14 |
| | | Low operating cost | 17.45% | 4.36% | 11 |
| | | Modularity easily adjustable | 9.84% | 2.46% | 16 |
| | | Multiple-lease potential | 8.62% | 2.15% | 17 |
| Designers' concerns | 24.83% | Firm's experience | 44.49% | 11.12% | 1 |
| | | Elegant presentation | 32.54% | 8.13% | 3 |
| | | Familiarization with laws and regulations | 12.69% | 3.17% | 12 |
| | | Excellent word-of-mouth | 10.25% | 2.56% | 15 |
| Environmental consciousness | 11.36% | Green building materials | 41.79% | 10.44% | 2 |
| | | Energy conservation | 29.77% | 7.44% | 6 |
| | | Water conservation | 28.43% | 7.10% | 7 |
| Total *Wi* | 100% | | | 100% | |

## 7. Discussion

*The Top Five Important Items*

The five most important elements were 1. firm's experience (11.12%), 2. green building materials (10.44%), 3. elegant presentation (8.13%), 4. adjacent unit connection (7.72%), and 5. cultural heritage connection (7.71%). The results of questionnaire analysis indicated that interior design service providers and shopping mall operators considered experience to be the most important element and this was followed by green building materials [23]. This indicated that design service providers and shopping mall operators value the concept of earth-friendly green buildings. Additionally, elegant presentations were used to understand the actual content of a proposal and its peripheral benefits [23]. Therefore, in addition to aesthetics, designers must use the actual conditions of shopping malls, adjacent unit connection [24], cultural heritage connection, and outstanding cases in other industries as a basis for conducting effective analysis and connection [25].

This enables designers to provide planning suggestions that are different from the traditional model and create other values for the shopping malls [26], thereby winning the hearts of business operators and the design project [27].This research has undergone case empirical research with AHP recognized by experts, and the relevant weights and objective quantitative evaluations obtained can actually prove that the cultural shopping mall indicators established in this research can provide sustainability, renovation, cultural shopping center space and space reuse provide an important reference.

## 8. Conclusions

Designers and operators must attempt to draw from different orientations, to improve cultural malls, and to appeal to the general visitors who are interested in non-architectural, but cultural, historical, and cultural assets [1,28]. If you want to analyze the cultural park or the topic of reuse, you must have at least a basic understanding of the space, form, and function of the building [29]. For the discussion and application of the cultural mall, it also involves the space of the building, the building's use, and thinking about the reuse of old buildings. [13,30]

On the other hand, different types of viewers expressed their views on architecture and cultural assets during the visit [2,31]. How is one to gradually promote a good concept to make more people understand and promote dialogue and communication between the people and the profession? That is to say, how will the preservation of architectural history, preservation of cultural assets [32] and the sustainability of the historical environment of the city be further implemented into the general concepts of the people, greatly increasing the willingness of ordinary visitors to enter the cultural mall.

We have proposed a method of implementing design decisions based on a comprehensive view of our Delphi method expert questionnaire and our AHP questionnaire [32]. Our method has exploratory and quantitative effectiveness, it reveals the importance and popular aspects of the spaces of cultural shopping malls [33].

Additionally, the key factors of cultural shopping mall spaces were revealed by analysis and interviews as well as case study analysis. Specifically, the effects of these factors on management strategy, management success, and development of spaces of culture were disclosed [34]. In addition to shopping mall services, numerous cultural shopping malls normally provide various fitness facilities and activities. Some shopping malls even provide various types of delicacies, high-end guestrooms, large wedding reception halls, and conference rooms, which make these shopping malls the main attraction spots of vacation resorts [29]. Moreover, in terms of the flow of spaces and convenience and efficiency of facilities, refinement of spatial equipment influenced the popularity of double-function cultural spaces in the market. Additionally, it created added value for customers and improved satisfaction of all customers to increase their loyalty. Therefore, the refinement is worthwhile for the development of cultural shopping mall spaces. Through investigation of cultural shopping mall spaces, this study identified, and measured the main factors in the refurbishment of cultural shopping mall spaces, thereby serving as a sustainable cultural shopping mall space management model.

*Research Limitations and Future Study*

Future research regarding operational costs must consider the combination of operational costs with complete practical experience. Moreover, the ability to integrate each item and comprehensiveness of understanding of materials are necessary. In reality, law and regulation problems directly influence future operational costs and overall design directions [35]. If one is unable to meet laws and regulations, any design is infeasible. Data collection was difficult because only a few cultural shopping malls exist in Taiwan and evaluation based on actual cases is necessary [36]. We recommend future research to study cultural shopping malls in Asia. This may enable Taiwan to earn international recognition in design and aesthetic application and make Taiwan a key place for knowledge management and creative design in Asia [37,38]. Furthermore, Taiwan may become the leader in the development of creative design for Chinese culture and expand the country's thriving design power to the international market through cultural and creative industries.

**Author Contributions:** Conceptualization & Formal analysis: M.-F.L., S.-G.S. and Y.-H.P.; Investigation: M.-F.L.; Resources: M.-F.L.; writing—original draft preparation: M.-F.L.; writing—review and editing: M.-F.L.; All authors have read and agreed to the published version of the manuscript.

**Funding:** This research received no external funding.

**Conflicts of Interest:** The authors declare no conflict of Interest.

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
