# Peer review of "Sustainable Shopping Mall Rehabilitation"

_sustainability, doi:10.3390/su12176698_

Round 1
Reviewer 1 Report
The abstract is not well structured. it is missing the purpose, problem, methods, and findings.
The article has no theoretical background.
The empirical research is not well explained.
Who are the participants in the Delphi study?
How were they selected?
The results need support on a theoretical framework.
Reviewer 2 Report
The paper is interesting and original, as it provides insights into the Sustainable Shopping Mall Rehabilitation.
Below I would give some comments for better clarify the research path and structure following MDPI sustainability template (https://www.mdpi.com/journal/sustainability/instructions):
1 Introduction
The introduction (aims and background) can’t start with “This study used an in-depth interview method to survey interior decoration project service providers and experts” because it is a part of materials and methods paragraph. The theoretical background isn’t suitable for the aim of the paper, I would suggest implementing some key references on sustainability, refurbishment, culture shopping mall space regeneration and multicriteria methods. I would also suggest to better explain the whole research path highlighting the research question and research structure (explaining the following paragraphs) after the literature review suggested.
2 Materials and Methods
Materials and methods could be better explained, I would suggest inserting a research methodology explanation specifying each research step able in responding to the research objectives (a graphic on research methodology approach could help). I would also suggest to better highlight the choice of two methods (Delphi and AHP) and how they are used for the whole research focus.
- Results
In this part, I suggest to better underline every result achieved in relation to the research objectives of the introduction and showing the indicators’ hierarchical framework for interior design project (research objective 2).
- Discussion and conclusions
Please focalize the discussion on the first research objective: “Discuss evaluation indicators of interior design projects for cultural business centers”.
Reviewer 3 Report
It was good to see a paper focusing on culture shopping mall space. However in my opinion this paper is not ready for publication. This paper needs to be re-written for better flow and linkages of the different sections. Currently this reads as disjointed. Your abstract could have been more detailed on the contribution to knowledge and the process of data collection.
The paper structure goes from introduction to methods used: but what about the theoretical background of this study? What about a in-depth literature review? Culture shopping mall space pattern research needs to be discussed. It would be good if the authors reinforced it in some way to connect the study with the objectives of this journal.
The description of the Methodology is lacking in detail and scope. The analysis of the data and the methodology is not shown as robust from the scientific point of view and is not well explained.
The discussion do not provide enough detail. Also, the discussion section could benefit from an engagement with the literature that addresses the novelty of the approach, the significance of integrating models and how it might contribute to broader knowledge in the field.
Your conclusion was quite limited as there was very little on the implications of the findings. Also here I am struggling to understand the contribution to knowledge as most of your findings coincide with what already exists in the literature. Overall, this paper needs a re-write for clarity, justification of the decisions made in this research and the contribution to knowledge.
Round 2
Reviewer 1 Report
The authors addressed to my concerns.
The article can be accepted.
Reviewer 2 Report
No comments
Reviewer 3 Report
Thank you very much for the detailed and well-explained revision of the document. I think it will be a good contribution to our Journal and I recommend to accept the paper.